

# High Resolution Tropospheric Carbon Monoxide Profiles Retrieved from CrIS and TROPOMI

**Dejian Fu[1], Kevin W. Bowman[1], Helen M. Worden[2], Vijay Natraj[1], John R. Worden[1], Shanshan Yu[1], Pepijn Veefkind[3,4], Ilse Aben[5], Jochen Landgraf[5], Lawrence Strow[6], Yong Han[7]**

[1]{NASA Jet Propulsion Laboratory, California Institute of Technology, Pasadena, California, United States of America}

[2]{National Center for Atmospheric Research, Boulder, Colorado, United States of America}

[3]{Royal Netherlands Meteorological Institute, Utrechtseweg, De Bilt, The Netherlands}

[4]{Delft University of Technology, Department of Geoscience and Remote Sensing, Stevinweg 1, Delft, The Netherlands}

[5]{SRON Netherlands Institute for Space Research, Sorbonnelaan, Utrecht, The Netherlands}

[6]{University of Maryland, Baltimore, Maryland, United States of America}

[7]{Center for Satellite Applications and Research, National Environmental Satellite, Data, and Information Service, NOAA, College Park, Maryland, United States of America}

Correspondence to: Dejian Fu (dejian.fu@jpl.nasa.gov)

**Abstract**

The Measurements of Pollution in the Troposphere (MOPITT) instrument is the only satellite-borne sensor in operation that uses both thermal (TIR) and near infrared (NIR) channels to estimate CO profiles. With more than fifteen years (2000 to present) of validated multi-spectral observations, this air quality and climate record provides the unique capability to separate CO in the Lower Most Troposphere (LMT, surface to 3 km (~700 hPa)) from the free tropospheric abundance. To extend this record, a new, hyper-spectral approach is presented here that will provide CO data products exceeding the capabilities of MOPITT instrument by





combining the short wavelength infrared (SWIR, equivalent to the MOPITT NIR) channels
from The TROPOspheric Monitoring Instrument (TROPOMI) aboard the European Sentinel 5
Precursor (S5p) satellite, and the TIR channels from the Cross-track Infrared Sounder (CrIS)
aboard Suomi National Polar-orbiting Partnership (Suomi NPP) satellite. We apply the MUlti-
SpEctra, MUlti-SpEcies, MUlti-SEnsors (MUSES) retrieval algorithm, to quantify the
potential of this joint CO product. CO profiles are retrieved from a single-footprint, full
spectral resolution CrIS transect over Africa on August 27-28, 2013 coincident with
significant biomass burning. Comparisons of collocated CrIS and MOPITT CO observations
for the LMT show a mean difference of 2.8±24.9 ppb, which is well within the estimated
measurement uncertainty of both sensors. The estimated degrees of freedom (DOFS) for CO
signals from synergistic CrIS/TROPOMI retrievals are 0.9 in the LMT and 1.3 above LMT,
which indicates that the LMT CO can be distinguished near surface CO abundance
information from the free troposphere, similar to MOPITT multiple spectral observations (0.8
in the LMT, and 1.1 above LMT). In addition to increased sensitivity, the combined retrievals
reduce measurement uncertainty with ~15% error reduction in LMT. With a daily global
coverage and a combined spatial footprint of 14 km, the joint CrIS/TROPOMI measurements
have the potential to extend and improve upon the MOPITT multi-spectral CO data records
for the coming decade.

## 20   1   Introduction

Observations of tropospheric carbon monoxide (CO) from space over the last decade have
been exploited for monitoring air quality (e.g., Clerbaux et al., 2008; Kar et al., 2010),
quantifying CO emissions (e.g., Kopacz et al., 2009; Fortems-Cheiney et al., 2011), analyzing
long-range transport of pollution (e.g., Heald et al., 2003; Edwards et al., 2006; Zhang et al.,
2006), attributing sources and sinks of $CO_2$ concentrations (e.g., Silva et al., 2013), and
evaluating chemical transport models and decadal trends in atmospheric composition (e.g.,
Shindell et al., 2006; Worden et al., 2013a). The Measurements Of Pollution In The
Troposphere (MOPITT) instrument, which is on the Earth Observation System EOS-Terra
platform, has acquired more than fifteen years (2000 to present) of validated global CO
observations (Emmons et al., 2007, 2009; Deeter et al., 2013, 2014). MOPITT is equipped
with gas filter correlation radiometers (Drummond, 1992) measuring both CO first
fundamental (4.6 μm) and overtone bands (2.3 μm). The synergy of CO first fundamental



band in the thermal infrared (TIR) and overtone band in the near infrared (NIR) provides an
unprecedented sensitivity to probe CO in the lowermost troposphere (LMT, surface to 3 km)
(Worden et al., 2010; Worden et al., 2013b). This unique multi-spectral capability of
MOPITT is not available from any single sensor on existing satellites that depend on a single
spectral band, e.g., AIRS (Atmospheric Infrared Sounder) on EOS-Aqua (McMillan et al.,
2005; Warner et al., 2007), TES (Tropospheric Emission Spectrometer) on EOS-Aura
(Rinsland et al., 2006), IASI (Infrared Atmospheric Sounding Interferometer) on METOP-
A/B (George et al., 2009), SCIAMACHY on Envisat (de Laat et al., 2007). Retrieval
sensitivity to the LMT is critical for the operational use of satellite data in air quality, climate,
and carbon applications, motivating the multi-spectral retrieval approach for a variety of
species including CO (Landgraf and Hasekamp 2007; Worden et al., 2010; Cuesta et al.,
2013; Fu et al., 2013; Worden et al., 2013a, 2013b).
All NASA space missions capable of measuring atmospheric CO concentrations have passed
their nominal lifetime by years (Table 1). The European Space Agency (ESA) Sentinel 5
precursor (S5p) TROPOspheric Monitoring Instrument (TROPOMI), which is expected to
launch in 2016 into an afternoon orbit behind the Suomi National Polar-orbiting Partnership
(Suomi-NPP) satellite, has only NIR channels for CO measurements. The Cross-track
Infrared Sounder (CrIS) aboard the Suomi-NPP satellite is a TIR sensor operating since
October 28, 2011 and providing observations at full spectral resolution since December 4,
2014. The constellation of Suomi NPP and ESA S5p provides a unique set of collocated
observations, which could extend the EOS-MOPITT multi-spectral CO data products with
significant improvements on spatial resolution and coverage (Table 1). Joint CrIS/TROPOMI
retrieval algorithm can also be applied to the future joint Sentinel-5 UVNS/IASI-NG
observations from METOP Second Generation satellites (Veefkind et al., 2012; Crevoisier et
al., 2014), which could provide joint TIR/NIR CO measurements in the time period of 2022-

26   2045.

In this paper, we describe the MUlti-SpEctra, MUlti-SpEcies, Multi-SEnsors (MUSES)
retrieval algorithm that combines the TROPOMI and CrIS spectral radiances to produce
atmospheric CO Volume Mixing Ratio (VMR) profiles with a vertical resolution that
improves upon the EOS-Terra MOPITT multi-spectral CO data products. This multi-spectral
observation strategy offers two significant advantages relative to traditional single band





measurements: enhanced sensitivity to composition changes especially in the LMT, and
reduced measurement uncertainty.
The paper is organized as follows: Section 2 describes the characteristics of CrIS/TROPOMI
measurements and the pairing strategy. Section 3 describes the MUSES retrieval algorithm,
samples of retrievals using CrIS full spectral resolution, single-footprint measurements,
comparisons of collocated CrIS and MOPITT observations and estimated characteristics of
synergistic CrIS/TROPOMI retrievals. Conclusions are presented in Section 4.

## 9    2    CrIS and TROPOMI

CrIS is on the Suomi NPP satellite in a near-polar, sun-synchronous, 828 km altitude orbit
with a 1:30 pm equator crossing time (ascending node), and has been operational since
October 28, 2011. TROPOMI will be on the ESA S5p satellite, planned for launch in summer
2016 with a design lifetime of 7 years (Table 1). S5p will fly within approximately 5 minutes
of Suomi NPP, which enables collocated observations of atmospheric composition (cloud,
aerosol, temperature, and trace gases) and surface properties (albedo, emissivity, and skin
temperature), thus building upon the success of the "A-Train" constellation of Earth
observation satellites.
CrIS is a Fourier transform spectrometer that measures the TIR radiances emitted by the
Earth's surface and transmitted through atmospheric gases and particles in three spectral
bands, including the long-wave IR band 1 (648.750-1096.250 $cm^{-1}$), the mid-wave IR band 2
(1208.750-1751.250 $cm^{-1}$), and the short-wave IR band 3 (2153.750-2551.250 $cm^{-1}$) (Han et
al., 2013, 2015; Strow et al., 2013a,b; Tobin et al., 2013; Wang et al., 2013). It was intended
as the operational successor to the AIRS instrument on the Aqua Platform (Aumann et al.,
2003; Pagano et al., 2003). The typical signal to noise ratio (SNR) of a CrIS measurement is
about 800:1 in the spectral region of interest for CO. CrIS is a cross-track scanning
instrument, whose full spectral resolution is 0.625 $cm^{-1}$, providing measurements with daily
global coverage (Table 1). Currently, the operational Level 1B products provide full spectral
resolution only for the long-wave IR band 1 for entire lifetime of the mission. The full
resolution (0.650 $cm^{-1}$) spectral radiance products for band 2 (was 1.250 $cm^{-1}$) and band 3
(was 2.500 $cm^{-1}$) have been available since December $4^{th}$, 2014. Ground pixels have a
diameter of 14.0 km at nadir. CrIS atmospheric measurements in the 2155-2209 $cm^{-1}$ spectral
region – nearly identical to observations of MOPITT TIR channels – include high-density





absorption features of the strongest fundamental band of CO and minor absorption from
interfering species, providing sensitivity for estimating the atmospheric CO concentration. It
is worth noting that Gambacorta et al., [2014] found that the information content present in
the CO retrievals improves up to one order of magnitude upon switching from spectral
resolution of 2.5 cm$^{-1}$ to the full spectral resolution of 0.625 cm$^{-1}$ (starting from December 4$^{th}$

6 2014).

TROPOMI will provide daily global coverage owing to its wide swath across track (Table 1).
It is a nadir-viewing push broom imaging spectrometer that measures backscattered and
reflected sunlight covering the 270-500 nm, 675-725 nm, 725-775 nm, and 2305-2385 nm
(4193-4338 cm$^{-1}$) spectral regions. Its atmospheric measurements in the 2.3 μm band – nearly
identical to observations of NIR channels of SCIAMACHY and MOPITT – include high-
density absorption features of the overtone band of CO, providing sensitivity for estimating
the CO total column average VMR. The module of the spectral band at 2.3 μm has a spectral
resolution of 0.25 nm and a spectral sampling rate of about 2.0-2.5 detector elements per
FWHM (Full Width at Half Maximum) (Veefkind et al., 2012). The ground pixel size of its
CO measurements at the nadir position is $7 \times 7$ km$^2$, which yields a spatial resolution about 10
times higher than Terra-MOPITT ($22 \times 22$ km$^2$) mission (Table 1). Within the spectral region
of interest, the minimum spectral SNR of a single TROPOMI measurement is 120:1 in the
continuum around 2310 nm (4329 cm$^{-1}$), specified for a scene with a surface albedo of 5%
and solar zenith angle of 70° (Veefkind et al., 2012).
In order to match the CrIS footprint size, our retrieval algorithm will average 4 adjacent pixels
of TROPOMI prior to the spectral fittings. Hence, the effective SNR of TROPOMI
measurements for the synergistic retrievals will be greater than 240:1. The joint
CrIS/TROPOMI spatial resolution will be $14 \times 14$ km$^2$ at the nadir position, about 2.5 times
higher than that of the MOPITT mission (Table 1).
Daytime ascending node CO retrievals are available from TROPOMI and CrIS whereas
nocturnal descending node CO depends exclusively on CrIS. The daily spatial sampling of the
joint CrIS/TROPOMI measurements is more than 8.0 times better than that of Terra MOPITT
since CrIS/TROPOMI measurements have ~4 times wider swath and 2.5 times finer ground
pixel size compared to MOPITT (Table 1). After performing the temporal and spatial matches
among CrIS/TROPOMI measurements, the distances between matched CrIS/TROPOMI
observations in the nadir direction are within 3 km – smaller than the pixel sizes of both





instruments. The associated temporal differences are within 5 minutes. In general, these
spatial and temporal separations are small compared to the scales of variability anticipated for
CO and could be neglected. Over complex source regions, such as urban areas, the special
treatments on achieving perfect spatial match might be necessary.

## 6    3   Retrieval algorithm, sample results and retrieval characteristics

This section describes the MUSES algorithm for synergistic retrievals of CO profiles (Section
3.1), sample retrievals when only using CrIS measurements (Section 3.2), synthetic joint
CrIS/TROPOMI CO retrievals to access the characteristics of improved tropospheric CO
profiling when combining TIR/NIR observations (Section 3.3).

### 11    3.1   MUSES retrieval algorithm for producing joint TROPOMI and CrIS carbon
### 12          monoxide volume mixing matrix profile data products

The MUSES retrieval algorithm is based upon Optimal Estimation (OE) method (Rodgers
2000). OE combines a priori knowledge, which includes both a mean state and covariance of
the atmospheric state, and the measurements to infer the atmospheric state. The OE algorithm
computes the best estimate state vector $\hat{\mathbf{x}}$, which represents the concentration of atmospheric
trace gases and ancillary parameters, by minimizing the following cost function

$$C(\mathbf{x}) = \|\mathbf{x} - \mathbf{x}_a\|^2_{\mathbf{S}_a^{-1}} + \|\mathbf{L}_{\mathrm{obs}} - \mathbf{L}_{\mathrm{sim}}(\mathbf{x})\|^2_{\mathbf{S}_\epsilon^{-1}} \quad (1)$$

Equation 1 is a sum of quadratic functions representing Euclidean $\mathbf{L}^2$ norm ($\|\mathbf{b}\|^2_\mathbf{A} = \mathbf{b}^\mathbf{T}\mathbf{A}\mathbf{b}$),
with the first term accounting for the difference between the retrieved vector $\mathbf{x}$ and a priori
state $\mathbf{x}_a$, inversely weighted by the a priori matrix $\mathbf{S}_a$, and with the second term representing
the difference between the observed $\mathbf{L}_{\mathrm{obs}}$ and simulated $\mathbf{L}_{\mathrm{sim}}(\mathbf{x})$ radiance spectra inversely
weighted by the measurement error covariance matrix $\mathbf{S}_\epsilon$.
Under the assumption that measurement error between TROPOMI and CrIS are uncorrelated,
Eq. 1 can be written as:

$$C(\mathbf{x}) = \|\mathbf{x} - \mathbf{x}_a\|^2_{\mathbf{S}_a^{-1}} + \underbrace{\left\|\mathbf{L}_{\mathrm{obs\_TROPOMI}} - \mathbf{L}_{\mathrm{sim\_TROPOMI}}(\mathbf{x})\right\|^2_{\mathbf{S}_{\epsilon\_\mathrm{TROPOMI}}^{-1}}}_{\mathrm{TROPOMI}}$$

$$+ \underbrace{\left\|\mathbf{L}_{\mathrm{obs\_CrIS}} - \mathbf{L}_{\mathrm{sim\_CrIS}}(\mathbf{x})\right\|^2_{\mathbf{S}_{\epsilon\_\mathrm{CrIS}}^{-1}}}_{\mathrm{CrIS}} \quad (2)$$





The joint retrieval algorithm iteratively updates the state vector based upon a trust-region
Levenberg-Marquardt optimization algorithm to minimize the cost function in Eq. 2 (Rodgers
2000; Bowman et al., 2006):

$$\mathbf{x}_{i+1} = \mathbf{x}_i + \left[ \mathbf{S}_a^{-1} + \underbrace{\mathbf{K}_{TROPOMI}^T \mathbf{S}_{\varepsilon\_TROPOMI}^{-1} \mathbf{K}_{TROPOMI}}_{TROPOMI} + \underbrace{\mathbf{K}_{CrIS}^T \mathbf{S}_{\varepsilon\_CrIS}^{-1} \mathbf{K}_{CrIS}}_{CrIS} \right]^{-1}$$

$$* \left[ \mathbf{S}_a^{-1}(\mathbf{x}_a - \mathbf{x}_i) + \underbrace{\mathbf{K}_{TROPOMI}^T \mathbf{S}_{\varepsilon\_TROPOMI}^{-1} \Delta \mathbf{L}_{TROPOMI}}_{TROPOMI} + \underbrace{\mathbf{K}_{CrIS}^T \mathbf{S}_{\varepsilon\_CrIS}^{-1} \Delta \mathbf{L}_{CrIS}}_{CrIS} \right] \quad (3)$$
where, **K** is the Jacobian matrix representing sensitivity of spectral radiances to the
atmospheric state; and $\Delta$**L** is the difference between observed and simulated spectral
radiances.
To simulate thermal infrared spectral radiances $\mathbf{L}_{sim\_CrIS}$ and Jacobians $\mathbf{K}_{CrIS}$, the joint
TROPOMI and CrIS algorithm incorporates the forward model of TES operational algorithm
(Bowman et al., 2006, Clough et al., 2006), with CrIS specifications (spectral range,
resolution, SNRs, and viewing geometry) obtained from CrIS L1B data products. In the NIR
region, we use the Vector Linearized Discrete Ordinate Radiative Transfer (VLIDORT)
model (Spurr, 2006, 2008), with the specification for TROPOMI measurements (spectral
range, resolution, SNRs) described in Veefkind (2012), to compute the spectral radiances
$\mathbf{L}_{sim\_TROPOMI}$ and Jacobians $\mathbf{K}_{TROPOMI}$. The characteristics of joint CrIS/TROPOMI CO
retrievals will be illustrated in section 3.3.
The joint TROPOMI and CrIS retrievals start with the list of the fitting parameters, a priori
values and a priori uncertainty shown in Table 2. In addition to the initial guess for the trace
gas concentration (CO, $H_2O$, $CH_4$, and $N_2O$), the initial guess for auxiliary parameters used in
the simulation of CrIS radiances (including temperature profile, surface temperature and
emissivity, cloud extinction and top pressure) are also retrieved from CrIS radiances in order
to take into account their spectral signatures in the CO spectral regions.
The tandem orbit of Suomi NPP and S5p enables access to the high spatial resolution cloud
information measured by sensors aboard Suomi NPP. This convoy enables cloud-
prescreening using Visible Infrared Imaging Radiometer Suite (VIIRS) cloud products
(Platnick et al., 2013), which is a scanning radiometer on the Suomi NPP satellite. This cloud
prescreening improves the efficiency of data processing and the quality of the retrieved



profiles. Cloudy scenes for the CrIS geometry (~14.0 km diameter at nadir) will be
characterized using the mature infrared cloud forward modeling techniques used in the TES
retrievals (Kulawik et al., 2006; Eldering et al., 2008). For the cloud and aerosol radiative
modeling within the Field Of View (FOV) of the TROPOMI sensor, we will adopt the
algorithm that has been used in the production of Orbiting Carbon Observatory 2 (OCO-2)
version 6 Level 2 standard products (Boesch et al., 2015). We will retrieve the Gaussian
parameters that represent the optical depth profiles for water/ice clouds and top two aerosols
specified by the Modern Era Retrospective analysis for Research and Applications aerosol
reanalysis (MERRAero) climatology (2009-2010) (Table 2). We will use VIIRS Level 2
products of surface temperature (Hook et al., 2012), and cloud property (cloud fraction, cloud
optical thickness, and cloud top pressure (Baker 2011a,b; 2012)) as a priori information to
retrieve ancillary parameters.
Optimal estimation theory provides a tool to enable estimation of the characteristics e.g.,
vertical resolution/sensitivity and uncertainty of measurements, and was applied to evaluating
the characteristics of Aura TES (Bowman et al., 2006), joint TES/OMI data products (Fu et
al., 2013), and MOPITT data products (Worden et al., 2010). The averaging kernel matrix ($\mathbf{A}$)
and error covariance ($\mathbf{S}$) of CO profiles, can be calculated using the following equations
(Rodgers, 2000),
$$\mathbf{A} = \mathbf{GK} \qquad (4)$$
and
$$\mathbf{S} = (\mathbf{I} - \mathbf{A})\mathbf{S}_a(\mathbf{I} - \mathbf{A})^\mathrm{T} + \mathbf{GS}_\varepsilon\mathbf{G}^\mathrm{T} \qquad (5)$$
Where, $\mathbf{I}$ is the unitary matrix, $\mathbf{S}_a$ is the a priori covariance matrix of retrieved sate, $\mathbf{S}_\epsilon$ is the
measurement noise covariance, and $\mathbf{G}$, the gain matrix, can be written as the following
equation,
$$\mathbf{G} = (\mathbf{K}^\mathrm{T}\mathbf{S}_\epsilon^{-1}\mathbf{K} + \mathbf{S}_a^{-1})^{-1}\mathbf{K}^\mathrm{T}\mathbf{S}_\epsilon^{-1} \qquad (6)$$
**3.2   Carbon monoxide measured from CrIS and MOPITT during an African**
**biomass burning event**
A biomass-burning event is observed in Aqua MODIS fire data products (Giglio et al., 2003;
Davies et al., 2004), collocated with the CrIS and MOPITT ground tracks, on August 27th,



28th 2014 over Africa (Fig. 1). The CO concentration in the LMT during this biomass-burning
event shows a strong latitudinal gradient with local variation, based on the MOPITT
observations. This gradient provides an excellent opportunity to evaluate the performance of
CrIS and future CrIS/TROPOMI retrievals (section 3.3). We applied the MUSES algorithm to
retrieve CO profiles using real CrIS full spectral resolution, single footprint measurements,
and then compared the retrieved profiles to the collocated MOPITT observations. When
running in the retrieval mode that only uses CrIS measurements, the TROPOMI terms in the
right-hand side of Eq. 2 and Eq. 3 vanish.
The CrIS measurements clearly capture the CO gradient centered at 10°S and diminishing
poleward to roughly 20°S (Figs. 2 and 3), associated with the biomass-burning event detected
by the enhanced MODIS radiative fire power (Fig. 1). Table 3 shows the mean and RMS of
the difference between CrIS and MOPITT TIR CO observations in LMT is -6.9±22.8 ppb,
respectively, which is better than the differences between CrIS and MOPITT multi-spectral
observations (-22.9±38.8 ppb, Table 3). The differences between CrIS and MOPITT observed
CO VMR could arise from the following three sources: [1] a priori CO profiles used in the
retrievals  (Figs 2D, 2E; Fig. 3B); [2] measurement sensitivity; [3] measurement date/time
(MOPITT local time 10:30 am on August 27th, CrIS local time 1:30 pm on August 28th,

18  2013).

The diurnal variation of MODIS fire radiative power and fire counts for the time period of
August 27 to 28, 2013, is weak. In order to evaluate the impacts of the sources [1] and [2], the
retrievals for the same set of soundings were recomputed using a constant CO a priori profile,
which is 100.0 ppb in LMT (Supplemental Material Figure 1 and representative of clean air
conditions. When using common a priori profiles, both MOPITT and CrIS observations still
present a latitudinal pattern [Suppl. Figures 1 and 2] similar to the one shown in Fig. 3, which
indicates the consistency of measurements from two TIR sensors is insensitive to the choice
of a priori. The differences between CrIS and MOPITT TIR improved to 2.8 ppb (Suppl.
Table 1), showing similar measurement characteristics (sensitivity, accuracy, and precisions).
The mean and RMS of the differences between CrIS and MOPITT mutli-spectral CO data
products are -23.6±37.6 ppb (Table 3), which is greater than the differences between CrIS and
MOPITT TIR and also greater than the estimated measurement uncertainty. This difference is
expected between CrIS and MOPITT multispectral sensitivity as quantified by the averaging
kernels, which represent the sensitivity of the CO retrieval to the true state, as shown in Fig.





4A and Fig. 4B for CrIS and MOPITT, respectively. The higher amplitude averaging kernels
for MOPITT multi-spectral observations in Fig. 4B quantify the enhanced near surface
sensitivity.

## 3.3  Characteristics of joint TROPOMI and CrIS CO profile observations

To access the characteristics of improved tropospheric CO profiling when combining
TIR/NIR observations, we computed the averaging kernels, degrees of freedom for signal
(DOFS, the trace of averaging kernels), and error covariance matrix for both synergistic and
each instrument alone observations. We used the CrIS viewing geometry for the simulation of
TROPOMI measurements since the tandem orbit of Suomi NPP and TROPOMI is very
similar.
Figures 4-6 and Table 4 show the retrieval diagnostics. We find that the tropospheric CO
profiles generated from joint CrIS/TROPOMI measurements have vertical resolution (Fig.
4D), sensitivity (mean total DOFS of 2.22), and error characteristics (magenta lines in Fig. 6B
and 6C) similar to the MOPITT joint NIR/TIR measurements (Fig. 4A, Fig. 6A; Table 4,
mean total DOFS of 1.88). The synergistic CrIS/TROPOMI observations clearly distinguish
the LMT (green lines in Fig.4D) and middle troposphere (magenta lines in Fig. 4D). The
synergistic CrIS/TROPOMI measurements showed significant improvements on the
sensitivity in comparisons to individual measurements from both CrIS and TROPOMI
missions (Fig. 4B/C; Fig. 5; Table 4). The latitudinal gradient of biomass burning intensity
(Fig. 2F, Fig. 3C, fire counts 0-250; max fire radiative power 0-800 mW), CO concentration
(Fig. 3 A, 100-400 ppb), and the change of DOFS (Fig. 5, CrIS 1-1.7; TROPOMI 1-1.5) show
similar pattern across the transect. The synergistic CrIS/TROPOMI, however, had higher
DOFS, generally above 2.0, because of its higher LMT sensitivity (Table 4, DOFS of 0.9 for
joint vs. 0.6 for CrIS, TROPOMI alone). And the sensitivity of MOPITT TIR and CrIS
measurements for LMT are approximately identical (0.56 vs. 0.62). The estimated sensitivity
of joint CrIS/TROPOMI measurements show improvements in comparisons to MOPITT
synergistic TIR/NIR observations, possibly arise from TROPOMI is a 'staring' instrument,
which does not have the issue of geophysical radiance error found by Deeter et al. (2011) for
MOPITT NIR sensor that is a 'dragging' instrument. It is worth to note that the minimum
spectral SNR of TROPOMI measurements were used in synthetic TROPOMI alone and joint
CrIS/TROPOMI measurements, the actual performances of joint CrIS/TROPOMI could be
even better than that shown in Figs. 4-6.





The total error consists of two terms (Equation 5): the first term represents the smoothing error and the second term is the measurement error. By incorporating radiances measured by two nadir-viewing instruments (Equation 3), the error characteristics of joint CrIS/TROPOMI tropospheric CO estimates can be substantially improved, in comparison with joint CrIS/TROPOMI and each instrument alone. Particularly, in the altitude range from Surface to 3 km (~700 hPa), we have seen the total uncertainty reduced from 30% (using measurements from each instrument alone) to 15% (joint TIR/NIR). This increased sensitivity and decrease in uncertainty are critical for evaluating the role of tropical fires (or pyro-convection) on the CO distribution (Fromm et al., 2005, 2006).

## 4   Conclusions

Based upon the MUSES synergistic retrieval algorithm, the combined CrIS/TROPOMI observations can extend and improve on the EOS-Terra MOPITT multi-spectral carbon monoxide profile data products with the higher vertical resolution and accuracy compared to any single nadir-viewing platform and over 2.5 times higher spatial sampling than MOPITT.

The MUSES algorithm has been applied to retrieve carbon monoxide volume mixing ratio profiles using full spectral resolution, single footprint CrIS measurements over Africa on August 28, 2013. The agreement of retrieved carbon monoxide volume mixing ratio in the lower most troposphere (surface to 3 km; ~700 hPa) between CrIS and MOPITT TIR data products is -6.9±22.8 ppb) when using different a priori profiles in retrievals; and 2.8±24.9 ppb) when for using common a priori in retrievals.

The simulated synergistic retrievals of CrIS/TROPOMI yields 0.9 degrees of freedom for CO signals in the LMT and 1.3 above LMT, distinguishing signals from the LMT and that above LMT in the troposphere, similar to that of MOPITT multiple spectral observation (DOFS 0.8 in the LMT, and 1.1 above LMT). In addition to increased sensitivity, the joint retrievals reduce measurement uncertainty, especially in the LMT, about 15% error reduction in the altitude range from surface to 3 km (~700 hPa). The potentials of synergistic CrIS/TROPOMI observations will be fully exploit using MUSES algorithm when TROPOMI Level 1B spectral radiances become available (estimated in late 2016). The validation of CO retrievals using aircraft in situ profiles will be accomplished in the near future.



By achieving information content that rivals the EOS-Terra MOPITT measurements,
synergistic CrIS/TROPOMI CO observations not only demonstrably enhances the scientific
value of S5p TROPOMI and Suomi NPP, but also extends the climate and tropospheric
records needed to continue NASA EOS science. Furthermore, the broad coverage of Suomi
NPP will provide global CO (a key tracer gas in the diagnostics of transport and chemical
reaction processes in the atmosphere) that compliments the NASA Decadal Survey GEO-
CAPE geostationary sounder (Fishman et al., 2012; http://geo-cape.larc.nasa.gov/). GEO-
CAPE is envisaged as a member of the Committee on Earth Observing Systems (CEOS) air
quality constellation, which includes the Korean GEMS (Bak et al., 2013), ESA Sentinel-4
(Ingmann et al., 2012; www.ceos.org/acc), and possibly Canadian PCW missions (Nassar et
al., 2014; http://www.asc-csa.gc.ca/eng/satellites/pcw/). The joint CrIS/TROPOMI retrieval
algorithm can also be applied to the future joint Sentinel-5 UVNS/IASI-NG observations
from METOP Second Generation satellites, which could provide joint NIR/TIR CO
measurements in the time period of 2022-2045 (Veefkind et al., 2012; Crevoisier et al., 2014).
The joint CrIS/TROPOMI CO profiles will enable the quantification of transport and
transformation of atmospheric composition in the domains unobserved by this constellation.
This combination of low-earth orbiting and geostationary space sounders would provide an
unprecedented atmospheric composition observing system needed to address long-term
scientific questions in climate, air quality, and atmospheric chemistry.
**Acknowledgements**
The authors thank Drs. Annmarie Eldering, Michael R. Gunson, Susan S. Kulawik, Karen
Cady-Pereira, Vivienne H. Payne, Bradley R. Pierce, and Stanley P. Sander for many helpful
discussions. Support from the NASA ROSE-2013 Atmospheric Composition: AURA Science
Team program (grant number: NNN13D455T) is gratefully acknowledged. Part of the
research was carried out at the Jet Propulsion Laboratory, California Institute of Technology,
under a contract with the National Aeronautics and Space Administration. Copyright 2015,
California Institute of Technology. Government sponsorship acknowledged.



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





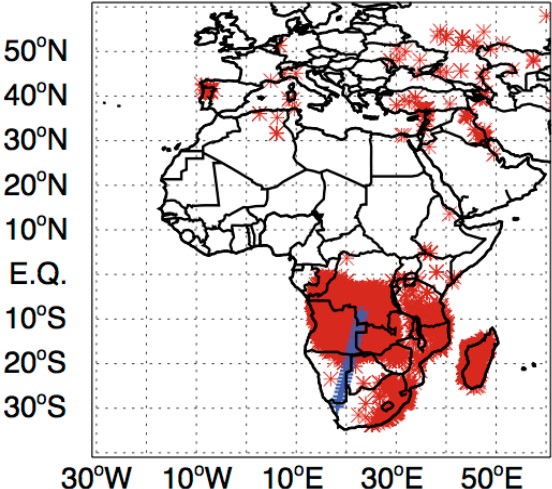

2    **Figure 1**. Collocated Suomi-NPP CrIS measurements (blue cross) over Africa on August 28[th],
3    2013 and Terra MOPITT observations on August 27[th], 2013. The red stars represent the fire
4    location measured by Aqua MODIS on August 28th, 2013.

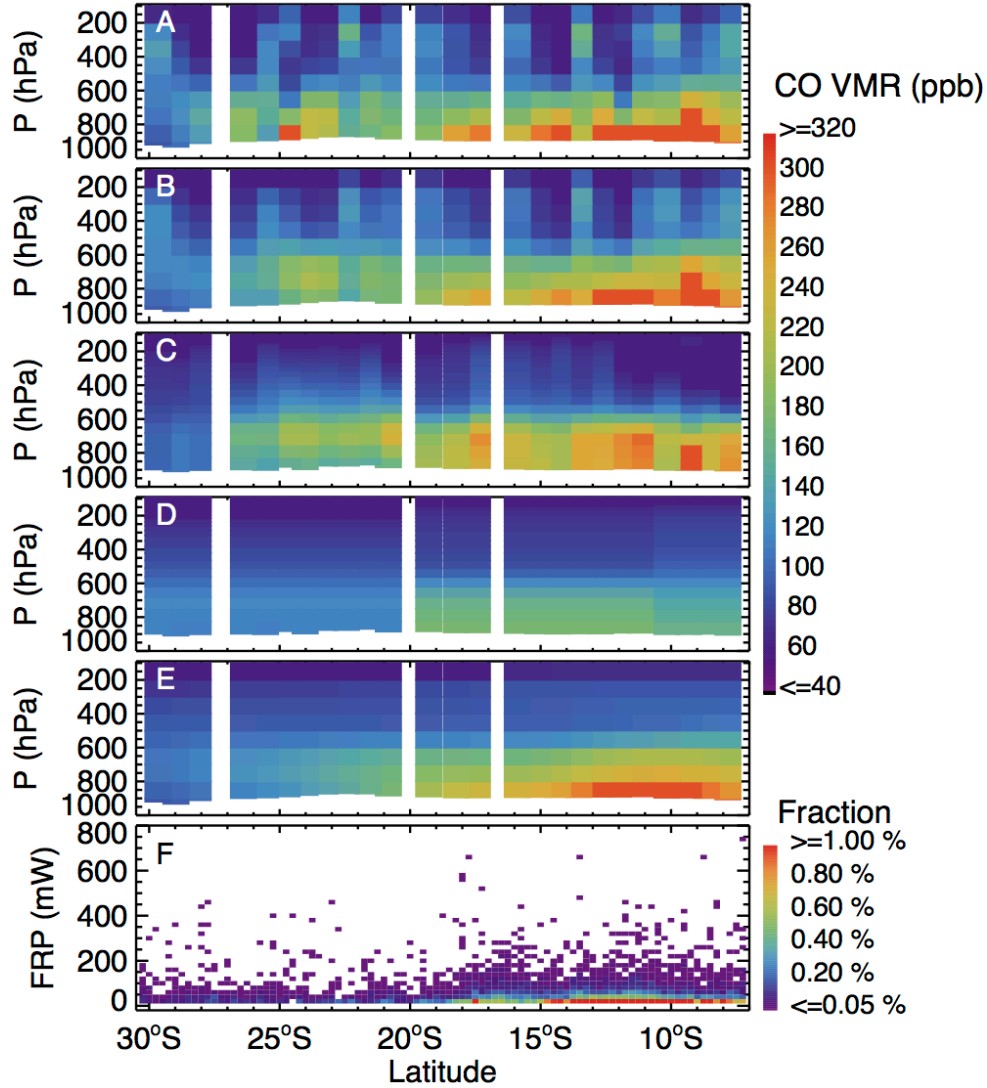

**Figure 2**. Tropospheric carbon monoxide (CO) volume mixing ratio (parts-per-billion)
profiles measured by Terra MOPITT (version 6.0) on August 27[th], Suomi-NPP CrIS on
August 28[th] 2013, and fire radiative power (milliwatts) measured by Aqua MODIS. (Panel A)
MOPITT multiple spectral CO fields; (Panel B) MOPITT thermal infrared CO fields; (Panel
C) CrIS thermal infrared CO fields using a priori profiles identical to those used in the Aura
TES operational retrievals; (Panel D) a priori CO fields used in CrIS retrievals shown in Panel
C. (Panel E) a priori profiles identical to those used in the MOPITT operational retrievals
shown in Panels A and B; (Panel F) Fire radiative power measured by Aqua MODIS over
Africa for August 28[th], 2013.




2

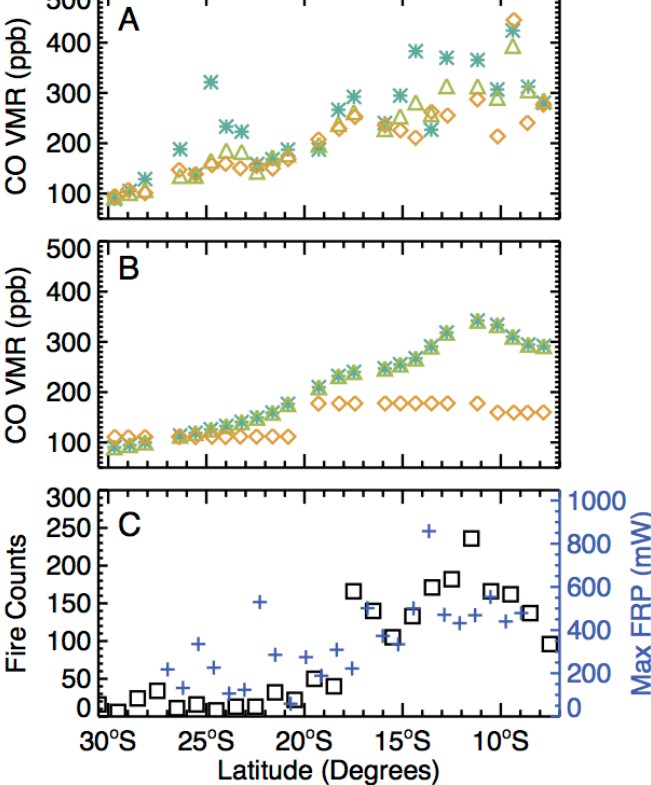

**Figure 3**. Averaged carbon monoxide (CO) volume mixing ratio (parts-per-billion) from surface to 3km (~700-hPa), fire counts and maximum fire radiative power (milliwatts) measured by Aqua MODIS over Africa for August 28[th], 2013. (Panel A) MOPITT multiple spectral CO data products (blue stars), MOPITT thermal infrared CO data products (green triangles), and CrIS thermal infrared CO VMR using a priori profiles identical to those used in the Aura TES operational retrievals (golden diamonds); (Panel B) a priori CO VMR used in MOPITT (green/blue) and CrIS (gold) retrievals. (Panel C) Fire counts (black squares) and maximum fire radiative power (blue plus) among the Auqa MODIS measurements whose data quality confidences are greater than 70%.






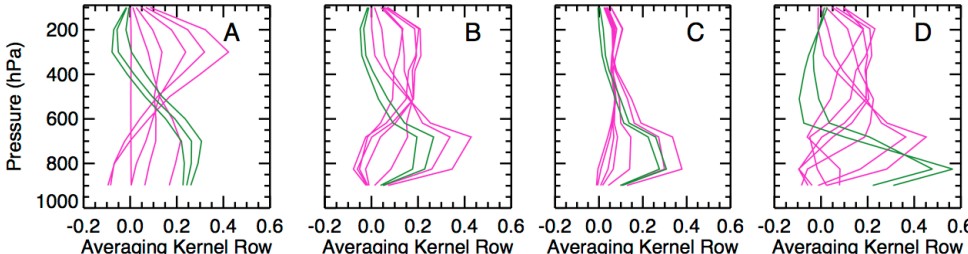

**Figure 4**. Sample averaging kernels of measurements for the target scene near 22.99°E,
8.65°S. In all panels, green lines are the averaging kernels from the surface to 3 km (~700
hPa); magenta lines are the averaging kernels from 3 km (~700 hPa) to 100 hPa. (Panel A)
MOPITT joint NIR/TIR measurements; (Panel B) Suomi-NPP CrIS measurements; (Panel C)
synthetic S5p TROPOMI measurements; (Panel D) synthetic joint TROPOMI/CrIS
measurements.





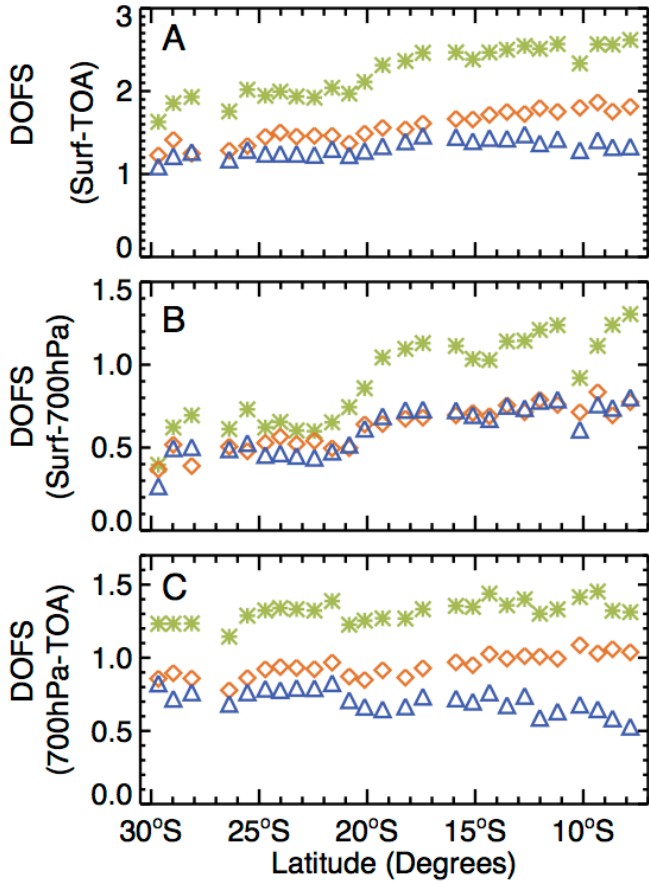

**Figure 5**. Degrees of freedom for CO measurements from CrIS, along with the synthetic
TROPOMI alone and joint CrIS and TROPOMI measurements over the biomass-burning
region. Green stars are the DOFS for joint TROPOMI and CrIS; gold diamonds are for the
DOFS for CrIS; and blue triangles are for DOFS from TROPOMI measurements.





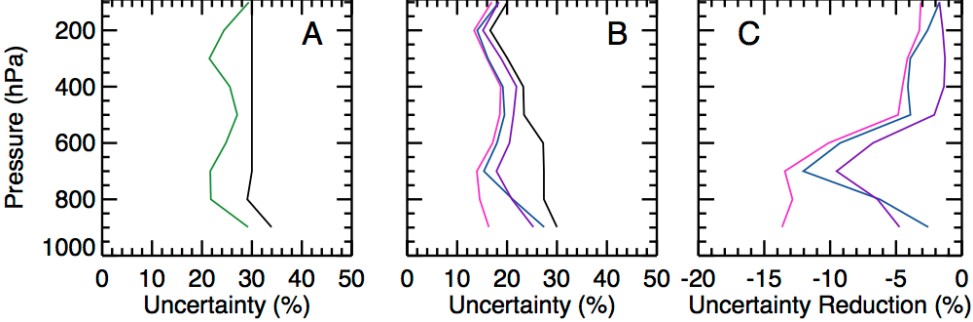

**Figure 6**. Uncertainties of the carbon monoxide (CO) volume mixing ratio (VMR) profiles
near 22.99°E, 8.65°S on August 28th, 2013. (Panel A) Uncertainty of the joint MOPITT
multiple spectral CO products (green line), and a priori profile (black line); (Panel B)
Uncertainty of CrIS actual CO measurements (blue line), synthetic TROPOMI (purple line)
and joint CrIS/TROPOMI (magenta line) CO measurements. Black line is the uncertainty of a
priori profile used in the retrievals; (Panel C) The reduction on the uncertainty with respect to
the uncertainty of a priori profile.



**Table 1**. Satellite missions that measure tropospheric carbon monoxide.

| Mission | Nominal Life Time | Years after Its Design Life Time | Spectral Resolution cm$^{-1}$ | | Footprint Size | Swath Width |
|---|---|---|---|---|---|---|
| | Start – End | Year | TIR[A] | NIR[B] | km$^2$ | km |
| CrIS/TROPOMI | 2016 – 2023 | 0 | 0.625[C] | 0.458 | 14 × 14 [D] | 2200 |
| MOPITT | 2000 – 2006 | 9 | ~0.04 eff [E] | ~0.25 eff [E] | 22 × 22 | 640 |
| CrIS | 2011 – 2026 | 0 | 0.625[C] | NA | $\pi \times 441$ [D,F] | 2200 |
| TES | 2004 – 2010 | 5 | 0.060[C] | NA | 8 × 5 | 5 |
| AIRS | 2002 – 2008 | 7 | ~ 1.800 | NA | $\pi \times 441$ [D,F] | 1600 |
| TROPOMI | 2016 – 2023 | 0 | NA | ~ 0.458 | 7 × 7 | 2600 |
| SCIAMACHY | 2002 – 2007 | Terminated [G] | NA | ~ 0.485 | 30 × 120 | 960 |

[A] First fundamental band of carbon monoxide, centered around 4.6 µm in the thermal infrared.
[B] First overtone band of carbon monoxide, centered around 2.3 µm in the near infrared.
[C] Specified values are the spectral resolution without appodization.
[D] The spatial resolution of data products from this work are 9 times higher than the existing operational CrIS and AIRS data products, since we use single footprint CrIS
6       L1B radiances in the retrievals, instead of cloud cleared radiances.
[E] MOPITT uses gas filter correlation radiometry (GFCR) with estimated effective spectral resolution.
[F] Estimated footprint sizes of AIRS and CrIS measurements since both sensors have circular fields of view.
[G] The measurements from SCIAMACHY ceased in 2012.



**Table 2.** List of parameters in state vector.

| Case Selection [A] | Fitting Parameters | Number of Parameters | A Priori | A Priori Uncertainty |
|---|---|---|---|---|
| CrIS/TROPOMI, CrIS, TROPOMI | CO at each pressure level | 14 | MOZART-3 [B] | MOZART-3 |
| CrIS/TROPOMI, CrIS, TROPOMI | $H_2O$ at each pressure level | 16 | GEOS-5 [C] | NCEP ~30% [D] |
| CrIS/TROPOMI, CrIS | $N_2O$ at each pressure level | 25 | MOZART-3 | MOZART-3 |
| CrIS/TROPOMI, CrIS, TROPOMI | $CH_4$ at each pressure level | 25 | MOZART-3 | MOZART-3 |
| CrIS/TROPOMI, CrIS | Surface temperature | 1 | GEOS-5 | 0.5K |
| CrIS/TROPOMI, CrIS | Surface emissivity [E] | 5 | UOW-M data base [F] | ~0.006 |
| CrIS/TROPOMI, CrIS | Cloud extinction [G] | 3 | Initial BT difference | 300% |
| CrIS/TROPOMI, CrIS | Cloud top pressure [G] | 1 | 500 mbar | 100% |
| CrIS/TROPOMI, TROPOMI | Gaussian parameters of optical depth profile for ice cloud [H] | 3 | [0.0125,0.30,0.04] [I] | [7.4,0.2,0.01] [I] |
| CrIS/TROPOMI, TROPOMI | Gaussian parameters of optical depth profile for water cloud [H] | 3 | [0.0125,0.75,0.10] [I] | [7.4,0.4,0.01] [I] |
| CrIS/TROPOMI, TROPOMI | Gaussian parameters of optical depth profile for primary aerosols [H] | 3 | [MERRA[J],0.90,0.05] [I] | [7.4,0.4,0.01] [I] |
| CrIS/TROPOMI, TROPOMI | Gaussian parameters of optical depth profile for secondary aerosols [H] | 3 | [MERRA[J],0.90,0.05] [I] | [7.4,0.4,0.01] [I] |
| CrIS/TROPOMI, TROPOMI | Surface albedo zero order term | 1 | from Spectra [K] | 0.2 |
| CrIS/TROPOMI, TROPOMI | Surface albedo first order term | 1 | 0 | 0.0005/cm$^{-1}$ |
| CrIS/TROPOMI, TROPOMI | Radiance/irradiance wavelength shifts | 2 | 0 | 0.5 cm$^{-1}$ |

[A] The parameters are included in the retrievals for different cases (CrIS only, TROPOMI only, and joint CrIS/TROPOMI)
[B] Model for OZone and Related chemical Tracers (MOZART) 3 (Brasseur et al. 1998; Park et al. 2004)
[C] Goddard Earth Observing System, version 5 (GEOS-5) (Rienecker et al., 2008)
[D] National Center for Environmental Prediction (NCEP) reanalysis (Kalnay et al., 1996)
[E] Retrievals over land, spectral surface emissivity is included.
[F] Global infrared land surface emissivity database at University of Wisconsin-Madison (UOW-M) (Seemann et al., 2008).
[G] For cloud treatment in TIR spectral region, we adopt the approach used in the TES Level 2 full physics retrieval algorithm (Kulawik et al., 2006; Eldering et al., 2008).
[H] For cloud treatment in NIR spectral region, we adopt the approach used in the OCO-2 Level 2 full physics retrieval algorithm (Pages 28-31, 44-45, Boesch et al., 2015).
The wavelength-dependent optical property would be scaled to that of TROPOMI.
[I] Gaussian parameters represent the total optical depth, peak altitude, and profile width. The peak altitude and profile width are normalized to the pressure at surface.
[J] Modern Era Retrospective analysis for Research and Applications aerosol reanalysis (MERRAero) climatology (2009-2010) (Rienecker et al., 2011; Buchard et al., 2015)
[K] The surface is assumed to be Lambertian with a variable slope in wavelength to the albedo, such that the albedo can vary linearly across the spectral band. A priori value
of surface albedo for zero order term are estimated from the measured continuum radiances, using the following equation: $A = \frac{\pi I}{\mu_0 I_0}$, where, I is the meuasred earth shine
radiance in the continuum, $I_0$ is the solar continuum spectral irradiance, $\mu_0$ is the cosine of the solar zenith angle.



1   **Table 3**. The differences of carbon monoxide volume mixing ratio in the lower troposphere (surface to 3km (~700 hPa)) between CrIS and
2   MOPITT measurements shown in Figure 3A.

| Data Product | Mean | RMS |
|---|---|---|
|  | ppb | ppb |
| CrIS - MOPITT TIR | -6.9 | 22.8 |
| CrIS - MOPITT Joint TIR/NIR | -22.9 | 38.8 |



1    **Table 4**. Degree of freedom for MOPITT, CrIS, and TROPOMI carbon monoxide measurements.

| Altitude Range | Sensor | TIR | NIR | Joint TIR/NIR |
|---|---|---|---|---|
| | MOPITT | 1.44 | 0.51 | 1.88 |
| Surface to Top of Atmosphere | CrIS | 1.57 | - | 2.22 |
| | TROPOMI | - | 1.32 | |
| | MOPITT | 0.56 | 0.30 | 0.77 |
| Surface to 3 km (~700 hPa) | CrIS | 0.62 | - | 0.91 |
| | TROPOMI | - | 0.61 | |
| | MOPITT | 0.89 | 0.21 | 1.11 |
| 3 km (~700 hPa) to Top of Atmosphere | CrIS | 0.94 | - | 1.32 |
| | TROPOMI | - | 0.71 | |

