# Peer review of "High Resolution Tropospheric Carbon Monoxide Profiles Retrieved from CrIS and TROPOMI"

_Atmospheric Measurement Techniques, 2015_

## Referee Comment (RC1) · Anonymous Referee #1 · 9 Mar 2016

This paper aims at presenting the potential of synergistic CrIS/TROPOMI retrievals using MUSES algorithm. CrIS is in operation whereas TROPOMI is to be launched in 2016. The study focuses on one day (August 28, 2013) over Africa. Even if the paper could have been improved with other case studies or comparisons with aircraft profiles, the paper is interesting and well written. The subject of the paper is appropriate to AMT. I found the paper clearly presented, well organized and useful for the community. I recommend the paper to be published after the authors have addressed the following minor comments:

General Comments:

About Section 3.2: 1) P 9 line 14: The authors should add the auxiliary data used in the retrieval process in the list of sources for the differences between CrIS and MOPITT.

[Figure]

The surface temperature, temperature profiles, emissivity, cloud information data used in CrIS and MOPITT retrieval algorithms are different and play an important role.

2) P 9 line 20: "to evaluate the impacts of the sources [1] and [2]": Only the source [1] is investigated when using a common a priori profile. P 9 line 23 -24: It should be said that when using a common a priori profile, both CrIS and MOPITT retrievals show smaller CO concentrations in the LMT. Rephrase (lines 24-25-26) "which indicates the consistency of measurements from two TIR sensors is insensitive to the choice of a priori". Suggestion: "which indicates that the measurement sensitivity does not depend on the choice of the a priori." The statement (lines 26-27) "The differences . . .precisions)." is a bit straightforward. It is not as simple. The differences are smaller when using a common a priori. It doesn't mean that the 2 instruments have the same characteristics. P 9 line 30: "the estimated measurement uncertainty": How much is it for MOPITT? CrIS?

The word "observations" should sometimes be replaced by "retrievals", "data" or "products". For instance in: "compared the retrieved profiles to the collocated MOPITT observations" (P9 line 6) "shows the mean and RMS of the difference between CrIS and MOPITT TIR CO observations in LMT" (P9 line 12) "When using common a priori profiles, both MOPITT and CrIS observations" "3.3 Characteristics of joint TROPOMI and CrIS CO profile observations" Etc.

Please remove all the unnecessary zeros (For instance, in Section 2: 648.750 > 648.75; 0.650 > 0.65; 14.0 > 14; 8.0 > 8 etc.)

In Figs. 2, 3, 4, 5, 6, S1, S2: It should be nice to have titles for the subplots if possible ("MOPITT TIR/NIR"; 'CrIS a priori" etc.). It would help the readers.

Comments:

P 2 line 2: Add "to be launched" and "in 2016" . . . Monitoring Instrument (TROPOMI) aboard the European Sentinel 5 Precursor (S5p) satellite > Monitoring Instrument

(TROPOMI) to be launched aboard the European Sentinel 5 Precursor (S5p) satellite in 2016

Introduction: The authors should mention the ACE-FTS CO product, which combined 2 bands (4.7 and 2.3 $\mu$m). Reference: Clerbaux et al., 2008.

P 3 line 8-12: This sentence should be rephrased. It looks like the references are about CO but they are not.

P 3 line 13: The authors are talking about the NASA space missions in the Introduction. In Table 1, we expect either the NASA satellite missions or all the "CO" missions. If you choose NASA missions, remove SCIAMACHY. If you choose all the missions, add ACE-FTS and IASI.

P 3 line 20: Add Gambacorta et al. (2014) after "2014".

P 3 line 27: Is it the first time that the MUSES algorithm is presented? Has it been developed for the joint CrIS/TROPOMI product? It should be mentioned. Is it based on an existing algorithm? Is there a reference?

P 4 line 12: It is said that TROPOMI's launch is planned for the summer. And it is said "late 2016" in the conclusion.

P 5 line 29: $\sim$4 times > $\sim$3.5 times

P 6 line 7: Recall "MUlti-SpEctra, MUlti-SpEcies, MUlti-Sensors" here.

P 7 line 18:"a priori uncertainty": Later in the paper, the authors are using "uncertainty" for error. Here I would suggest you use "variance".

P 8 line 14-15-16: "and was applied to evaluating the characteristics of Aura TES (Bowman et al., 2006), joint TES/OMI data products (Fu et al., 2013), and MOPITT data products (Worden et al., 2010)" should be removed, it does not bring anything here.
P 8 line 22: matrix of retrieved > matrix of the atmospheric

P 9 line 11: I would add "(or standard deviation)": RMS > RMS (or standard deviation)

P 9 line 12 and line 14: It would be nice to have the figures in % too.

P 9 line 30: "the estimated measurement uncertainty": How much is it for MOPITT? CrIS?

P 10 line 8-9: How have the TROPOMI measurements been simulated? With what code?

P 10 line 11-13: The authors are talking about the "tropospheric CO profiles" and then about "total DOFS". Suggestion to remove "tropospheric".

P 10 line 13: About the "error characteristics": They are not similar but smaller for CrIS/TROPOMI than for MOPITT NIR/TIR products.

P 10 line 21: the change of DOFS > DOFS (?)

P 11 line 1: The total error consists of > The total error (or uncertainty) consists of

P 11 line 19-20: between CrIS and MOPITT TIR data products is > between CrIS and MOPITT TIR data products –August 27th– is

P 11 line 29: "late 2016" See Comment P 4 line 12

P 28 Table 1: Satellite missions > NASA satellite missions (or add ACE-FTS and IASI, see comment P 3 line 13)

P 29 Table 2 A priori Uncertainty > A priori Variance

P 31 Table 4: Suggestion to put the figures "2.22", "0.91" and "1.32" in italics and mention in the caption that it is the synergistic CrIS/TROPOMI product.

Trivia:

P 2 line 2: . . .The TROPOspheric. . . > the TROPOspheric

P 3 line 2: lowermost troposphere > Lower Most Troposphere

P 3 line 7-8: METOP-A/B > METOP-A and B

P 8 line 22: sate > state

P 9 line 1: 2014 > 2013

P 9 line 3: observations. > data (Fig. 2A)

P 9 line 10: (Figs. 2 and 3) > (Figs. 2C, 2D and 3A)

P 9 line 16: Figs 2D > Figs. 2D

P 9 line 22: (Supplemental Material Figure 1 > (Supplemental Material Figure 1)

P 10 line 13: (mean total DOFS of 2.22) > (mean total DOFS of 2.22, Table 4)

P 10 line 16: Fig.4D > Fig. 4D

P 10 line 17: showed > show

P 10 line 22: had > has

P 10 line 23: generally above 2.0 > generally above 2 (Fig. 5A)

P 15 line 10: Amsterdam , 1992 > Amsterdam, 1992 (remove space after Amsterdam)

P 16 Han et al. 2013 and 2015 are not in the text.

Figure 2: P 23 line 5: MOPITT multiple spectral CO fields > MOPITT multiple spectral (TIR/NIR) CO fields or MOPITT joint TIR/NIR CO fields (like in the caption of Supplemental Material Figure 2)

Figure 2: P 23 line 5: MOPITT thermal infrared CO fields > MOPITT thermal infrared (TIR) CO fields or MOPITT TIR CO fields

Figure 2: P 23 line 8: identical to those used in the MOPITT > used in the MOPITT

Figure 3: P 24 line 6: ∼700-hPa > ∼700 hPa

Figure 3: P 24 line 12: Auqa > Aqua

Figure 3: same thing as comment Figure 2: P 23 line 5 and comment

Figure 2: P 23 line 5

Figure 4: the averaging kernels of MOPITT TIR/NIR and CrIS have been switched.

P 29 Table 2 : lines 2,3,4,5: The dots at the end of the sentences A, B, C D are missing. Line 3 : Brasseur et al. 1998 > Brasseur et al., 1998 Line 3 : Park et al. 2004 > Park et al., 2004 Line 14 : meuasred > measured Line 14 : earth > Earth

P 31 Table 4: Degree > Degrees Surface to 3 km > LMT: Surface to 3 km

Supplement Material Figure 2: line 3: 700-hPa > 700 hPa Line 8: Auqa > Aqua

Ref: C. Clerbaux, M. George, S. Turquety, K. A. Walker, B. Barret, P. Bernath, C. Boone, T. Borsdorff, J. P. Cammas, V. Catoire, M. Coffey, P.-F. Coheur, M. Deeter, M. De Mazière, J. Drummond, P. Duchatelet, E. Dupuy, R. de Zafra, F. Eddounia, D. P. Edwards, L. Emmons, B. Funke, J. Gille, D. W. T. Griffith, J. Hannigan, F. Hase, M. Höpfner, N. Jones, A. Kagawa, Y. Kasai, I. Kramer, E. Le Flochmoën, N. J. Livesey, M. López-Puertas, M. Luo, E. Mahieu, D. Murtagh, P. Nédélec, A. Pazmino, H. Pumphrey, P. Ricaud, C. P. Rinsland, C. Robert, M. Schneider, C. Senten, G. Stiller, A. Strandberg, K. Strong, R. Sussmann, V. Thouret, J. Urban, and A. Wiacek, Atmos. Chem. Phys., 8, 2569-2594, 2008.

---

## Referee Comment (RC2) · Anonymous Referee #2 · 14 Mar 2016

Review of High Resolution Tropospheric Carbon 1 Monoxide Profiles Retrieved from CrIS and TROPOMI by Fu et al.

This manuscript describes how multispectral retrievals of CO can be obtained outside of MOPITT's single instrument capabilities in a TROPOMI – CrIS joint retrieval. It's laudable that the authors undertook this important task, since as they point out MOPITT (and all other EOS instruments) are well past their expected lifetime, and it's important to look for alternatives. They make a good case for combining TROPOMI and CrIS, but it's unfortunate that they have to (?) use the one test day from 2013 to demonstrate the utility of their instrument. It would have been preferable, though perhaps not possible, to focus on more recent CrIS CO data since full resolution data has been available for over a year now. Nonetheless, the work is important and needs to be demonstrated first, as it is in this manuscript.

[Figure]

Combining the data from two instruments yields a small mismatch of about 3km and 5 min, smaller than the instrument foot print and adequate for all but challenging topography such as urban areas.

The presentation of the retrieval technique, including associated equations, is straightforward and largely based on authors' previous experience with TES. An application of the retrieval algorithm to an African biomass burning event is quite appropriate and informative. The results are quite encouraging, especially since they are an apparent improvement on what can be obtained from MOPITT combined retrieval. It will be good to test MUSES product in practice with real data from both instruments, and that can't be done soon enough. Given the operational nature of the two instruments used, MUSES seems to be not only providing a good product, but it is a good long term investment of research effort.

It would be good to talk about the loss of data due to clouds that can be expected from the MUSES retrieval.

P12, line 6: should be "complements" not "compliments"
* * *

---

## Author Comment (AC1) · 13 Apr 2016

Dear Referee#1,

Thank you for reviewing the manuscript and providing the comments. Please see our responses to your comments in the attached supplement zip file. The revised manuscript is included in the attached supplement zip. Thank you.

Sincerely yours, Dejian Fu

Please also note the supplement to this comment:
http://www.atmos-meas-tech-discuss.net/amt-2015-404/amt-2015-404-AC1-supplement.zip